# Block-wise Separable Convolutions: An Alternative Way to Factorize Standard Convolutions

## Abstract

Convolutional neural networks (CNNs) have demonstrated great capability of solving various computer vision tasks with nice prediction performance. Nevertheless, the higher accuracy often comes with an increasing number of model parameters and large computational cost. This raises challenges in deploying them in resource-limited devices. In this paper, we introduce block-wise separable convolutions (BlkSConv) to replace the standard convolutions in order to compress deep CNN models. First, BlkSConv expresses the standard convolutional kernel as an ordered set of block vectors each of which is a linear combination of fixed basis block vectors. Then it eliminates most basis block vectors and their corresponding coefficients to obtain an approximated convolutional kernel. Moreover, the proposed BlkSConv operation can be efficiently realized via a combination of pointwise and group-wise convolutions. Thus the constructed networks have smaller model size and fewer multiply-adds operations while keeping comparable prediction accuracy. However, it is unknown how to search a qualified hyperparameter setting of the block depth and number of basis block vectors. To address this problem, we develop a hyperparameter search framework based on principal component analysis (PCA) to help determine these two hyperparameters such that the corresponding network achieves nice prediction performance while simultaneously satisfying the constraints of model size and model efficiency. Experimental results demonstrate the prediction performance of constructed BlkSConv-based CNNs where several convolutional layers are replaced by BlkSConv layers suggested by the proposed PCA-based hyperparameter search algorithm. Our results show that BlkSConv-based CNNs achieve competitive performance compared with the standard convolutional models for the datasets including ImageNet, CIFAR-10/100, Stanford Dogs, and Oxford Flowers.

## 1 Introduction

In the past decade, Deep Learning (DL) has been the basis of many successes in artificial intelligence, including a variety of applications in computer vision, reinforcement learning, and natural language processing. One of the most popular deep neural networks is Convolutional Neural Network (CNN). With the help of various techniques such as residual connections and batch normalization, it is easy to train deep CNNs with many layers on powerful GPUs. While large-scale CNN models have achieved great successes, they require huge computational complexity and massive storage. For example, VGG16 (27) has 138 million parameters and requires 154700 million multiply-add operations (MAdds) to classify an image. It is a great challenge to deploy them in real-time applications, especially on devices with limited resources such as mobile phones and embedded systems. Thus, the prediction models are required be compact and fast while keeping acceptable accuracy. The main approach to be compact is the model compression which aims at establishing a tradeoff between

model efficiency and accuracy. In the area of model compression, methods to construct efficient and compact CNNs are mainly divided into two approaches: one approach is to compress trained CNNs and the other approach is to design new compact CNNs and train them from scratch. Many works based on the first approach suggested several techniques such as quantization (33), model pruning (6; 24), Huffman coding (6), and low rank factorization (12).

Studies in the second approach mainly explored many ways for factorizing convolutions. For instance, Szegedy et al. (30) improved GoogLeNet (29) through factorizing convolutions with larger spatial filters by a two-layer convolutional architecture with smaller spatial filters. At present, most factorizing methods are usually performed via a combination of depthwise convolution, pointwise convolution, and groupwise convolution. For example, in (25), the depth-wise separable convolutions (DSCs) were proposed where the standard convolution is decomposed into a depth-wise convolution and a pointwise convolution. The ShuffleNets (36; 18) utilizes pointwise group convolution with channel shuffle to decompose the standard convolution. Moreover, many lightweight models based on DSCs or groupwise convolutions such as MobileNets (8; 23; 7) and ShuffleNets (36; 18) were proposed to greatly reduce computation cost while maintaining accuracy.

In this paper, we follow the research path of the second approach and propose block-wise separable convolutions (*BlkSConv*) to replace standard convolutions. BlkSConv approximates a standard convolution as follows. A standard $k \times k \times M$ convolutional kernel can be represented as an ordered set of block vectors of size $k \times k \times t$. Since each block vector can be written as a linear combination of $k^2 t$ basis vectors of size $k \times k \times t$, this standard convolutional kernel can be viewed as an ordered set of block vectors each of which is a linear combination of $k^2 t$ basis block vectors. Then BlkSConv eliminates most basis block vectors and their corresponding coefficients to obtain an approximated convolutional kernel. As shown on the left of Figure 1, the extreme version of BlkSConv is called the basic BlkSConv where only one basis block vector is used. When carefully setting the depth of the block vector, that is the parameter $t$, an approximated convolution of fewer parameters can be obtained and the corresponding compact CNN has acceptable prediction performance compared to the standard convolutions. To increase the prediction accuracy of the basic BlkSConv, an enhanced version is proposed by increasing the number of basis block vectors, that is the parameter $s$, as shown on the right of Figure 1. However, adding too many basis block vectors will significantly increase the model size and computational cost. Thus there is a tradeoff between model efficiency/size and accuracy. To realize the full potential of the enhanced BlkSConv in trading-off model efficiency/size and accuracy, we propose a framework based on the principal component analysis to search for the hyperparameters $t$ and $s$ of each BlkSConv layer for the given standard convolutional network. The proposed search framework suggests a possible setting of parameters $t$ and $s$ such that the constructed model based on these selected hyperparameters may achieve high prediction accuracy while simultaneously satisfying the constraints of model size and model efficiency in terms of MAdds.

To summarize, our main contributions are as follows. First, we develop a new convolutional layer called *BlkSConv* to approximate the standard convolutional layer. To approximate a standard convolutional kernel, BlkSConv divides the kernel into blocks and approximates each block by a linear combination of several fixed basis block vectors. The constructed networks have small model size and cost fewer multiply-adds operations while maintaining acceptable prediction accuracy. Then, we also develop a search framework to determine the block depth and the number of basis block vectors such that the corresponding networks with selected hyperparameters achieve comparable prediction performance while simultaneously satisfying the constraints of model size and model efficiency. We also present experimental results to demonstrate the performance of selected BlkSConv-based CNNs based on our proposed hyperparameter search algorithm. Our results show that selected BlkSConv-based CNNs achieve competitive performance compared with the standard convolutional models for the datasets including ImageNet, CIFAR-10/100, Stanford Dogs, and Oxford Flowers.

## 2 Related Work

Many efforts have been devoted to improve the efficiency of CNNs which could be roughly divided into three categories. First, model pruning is a popular method to improve efficiency of CNNs. In (6; 37), their methods remove redundancy in the trained CNN model by pruning connection. In (6; 21; 20; 35), the calculation amount of the trained model is compressed via quantization. In (17; 11; 16; 9; 28), model filters that have small contributions are removed and the corresponding trained model is fine-tuned to preserve the performance.

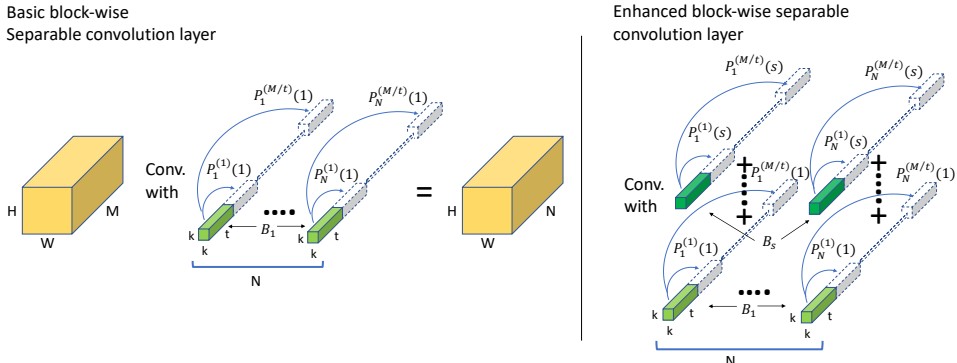

Figure 1: The proposed block-wise separable convolution and its enhanced version.

Second, many techniques are developed to factorize the standard convolutions. In (30), convolutions with larger spatial filters are factorized into two-layer convolutional architectures with smaller spatial filters. Through different combinations of depthwise convolution, pointwise convolution, and groupwise convolution, many well-known factorizing frameworks were developed. In (25), the depth-wise separable convolutions (DSCs) were proposed where the standard convolution is decomposed into a depth-wise convolution and a pointwise convolution. The ShuffleNets (36; 18) uses pointwise group convolution with channel shuffle to decompose the standard convolution. Moreover, many lightweight models based on DSCs or groupwise convolutions such as MobileNets (8; 23; 7) and ShuffleNets (36; 18) were proposed to greatly reduce computation cost while maintaining accuracy.

Recently, neural architecture search-based methods (34; 32; 38; 39; 31) have been proposed to automatically construct network architectures. These methods search over a set of network hyperparameters including different types of convolutional layers and kernel sizes, to find a network structure which satisfies optimization constraints such as inference speed. Major search frameworks include genetic-based methods (34) and reinforcement learning based methods (38). These techniques were used in state-of-the-art CNN architectures such as MnasNet (31) and MobileNetV3 (7).

Convolution weights of trained CNNs are also analyzed in (1; 3; 26; 4). Following their analysis, several approaches toward reducing redundant weights were proposed. In (2; 12; 13), the convolutional kernels are approximated via low-rank factorization. In (4), the kernels are analyzed via principal component analysis.

## 3 Block-wise Separable Convolutions (BlkSConv)

For any natural number $n$, let $[n]$ denote the set $\{1, 2, \ldots, n\}$. In a standard CNN, each convolutional layer converts an input tensor $I$ of size $M \times X \times Y$ into an output tensor $O$ of size $N \times X \times Y$ by applying the filter kernels $F_1, F_2, \ldots, F_N$, each of size $M \times \ell \times \ell$ with odd $\ell$ such that, for any $x, y, j \in [X] \times [Y] \times [N]$,

$$O(x, y, j) \;=\; \sum_{s_1=-(\ell-1)/2}^{(\ell-1)/2} \sum_{s_2=-(\ell-1)/2}^{(\ell-1)/2} \sum_{s_3=1}^{M} I(x + s_1, y + s_2, s_3) \cdot F_j(s_1, s_2, s_3). \tag{1}$$

During training, the weights of each kernel $F_j$ are optimized via backpropagation. The total number of weight parameters to be optimized in each kernel $F_j$ is $\ell^2 \cdot M$. In the subsequent work, we propose a framework to reduce the number of parameters of the standard convolutions while preserving its prediction performance. Then, in order to implement our new framework, we adopt a combination of pointwise and group-wise convolutions to efficiently realize the reduced convolutions. Combining these ideas, we introduce block-wise separable convolutions, denoted by *BlkSConv*. However, to generate a BlkSConv-based models, many hyperparameters should be determined for keeping prediction performance, model size, and model efficiency. Thus, we also propose an efficient hyperparameter search algorithm to select hyperparameters satisfying the given model constraints.

## 3.1 Expressing a standard convolution via a linear combination of block vectors

In this section, we propose block-wise separable convolutions. First, each convolutional kernel $F_j$ of size $M \times \ell \times \ell$ can be expressed as a concatenation of $M/t$ blocks $Q_j^{(1)}, Q_j^{(2)}, \ldots, Q_j^{(M/t)}$ each of size $\ell \times \ell \times t$ where $Q_j^{(k)}(x, y, z) = F_j(x, y, z + (k-1)t)$ for any $x, y, z \in [X] \times [Y] \times [t]$. We call $t$ the block depth. Let $\{B_1, B_2, \ldots, B_{t\ell^2}\}$ be a set of basis block vectors. Each $Q_j^{(k)}$ can be expressed uniquely as a linear combination of $B_1, B_2, \ldots, B_{t\ell^2}$, that is, there exist $t\ell^2$ values $P_j^{(k)}(i) \in \mathbb{R}$ such that $Q_j^{(k)} = \sum_{i=1}^{t\ell^2} P_j^{(k)}(i) \cdot B_i$. In practice, $t\ell^2$ may be large. In order to reduce the model size, we require the number of basis block vectors is fewer than or equal to a fixed number $s$ with $s < t\ell^2$. Now each $Q_j^{(k)}$ is replaced by the following linear combination of $B_1, \ldots, B_s$, that is $\widehat{Q}_j^{(k)} = \sum_{i=1}^{s} P_j^{(k)}(i) \cdot B_i$. The corresponding convolutional kernel $\widehat{F}_j$ is the concatenation of $M/t$ blocks $\widehat{Q}_j^{(1)}, \ldots, \widehat{Q}_j^{(M/t)}$. Therefore, the corresponding output tensor is

$$\widehat{O}(x, y, j) = \sum_{s_1, s_2 = -(\ell-1)/2}^{(\ell-1)/2} \sum_{s_3=1}^{M} I(x + s_1, y + s_2, s_3) \cdot \widehat{F}_j(s_1, s_2, s_3). \tag{2}$$

By Equation 2, the number of weight parameters in BlkSConv is $s \cdot (t \cdot \ell^2 + \frac{M}{t})$. To significantly reduce model size, we set $s = 1$. Figure 1 left illustrates the operation of BlkSConv when $s = 1$. In order to achieve the minimal model size, $t$ can be set as $\sqrt{M}/\ell$ and the number of parameters becomes $2\ell\sqrt{M}$ while the parameter number of the standard and $1 \times 1$ pointwise convolutions are $M\ell^2$ and $M$, respectively. Thus, the constructed BlkSConv-based CNNs have smaller model size than existing lightweight CNN models. Take the ResNet34 (10) as an example where, in the last stage of the ResNet-34, the convolutional kernel size is $3 \times 3$ and the channel size is 512, that is $\ell = 3$ and $M = 512$. In this case, the ratio between the parameter size of the BlkSConv-based convolutions and the standard convolutions is approximately 0.0295.

However, the prediction performance of the BlkSConv-based CNN with the smallest model size is usually worse than the standard CNNs. To increase accuracy, the number of basis block vectors should be increased, that is $s > 1$. Figure 1 right illustrates the operation of BlkSConv when $s > 1$. In this case, the number of parameters becomes $2s\ell\sqrt{M}$. Let us take convolutions in the last stage of ResNet-34 as examples. Let us set $t = 4$ in the BlkSConv. Now the ratio between the parameter size of the BlkSConv-based convolutions and the standard convolutions is approximately 0.0356. Thus we can add at least 5 basis block vectors to increase prediction accuracy. In this case, the ratio between the parameter size of the BlkSConv-based convolutions with 5 basis block vectors and the standard convolutions is approximately 0.178. In the experimental section, we demonstrate that the BlkSConv-based convolutions with few basis block vectors have prediction performance as well as the standard convolutions on ImageNet or even outperform the standard convolutions on several datasets when the backbone CNNs are ResNets.

The next problem is the computational efficiency of BlkSConv. If we compute the kernel $\widehat{F}_j$ first and perform a regular convolution according to the kernel $\widehat{F}_j$, then it is obvious that the computational cost is larger than the cost for just performing a standard convolution. We will address this problem in the subsequent section.

## 3.2 Implementation of BlkSConv via a combination of pointwise and group-wise convolutions

In this section, we propose an efficient implementation method to realize BlkSConv. The flowchart of the proposed implementation is illustrated in Figure 2. To derive an efficient implementation for BlkSConv operation, we rewrite Equation 2 as follows.

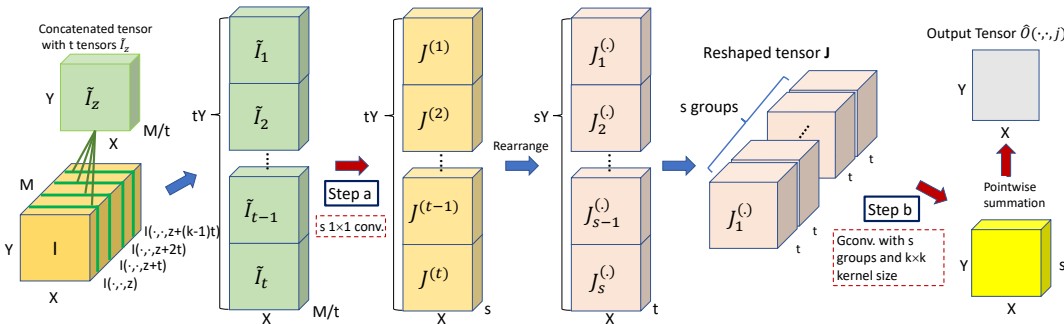

Figure 2: Flowchart of the block-wise separable convolution where Gconv. means the group-wise convolution.

$$\widehat{O}(x,y,j) = \sum_{s_1,s_2}\sum_{z=1}^{t}\sum_{k=1}^{M/t} I(x+s_1,y+s_2,z+(k-1)t)\cdot\widehat{Q}_j^{(k)}(s_1,s_2,z) \tag{3}$$

$$= \sum_{s_1,s_2}\sum_{z=1}^{t}\sum_{k=1}^{M/t} I(x+s_1,y+s_2,z+(k-1)t)\cdot\sum_{i=1}^{s} P_j^{(k)}(i)\cdot B_i(s_1,s_2,z) \tag{4}$$

$$= \sum_{i=1}^{s}\sum_{s_1,s_2}\sum_{z=1}^{t} B_i(s_1,s_2,z)\underbrace{\sum_{k=1}^{M/t} P_j^{(k)}(i)\cdot\overbrace{I(x+s_1,y+s_2,z+(k-1)t)}^{\widetilde{I}_z(x+s_1,y+s_2,k)}}_{J^{(z)}(x,y,i):\ \text{a point-wise convolution of } \widetilde{I}_z}. \tag{5}$$

Let $\widetilde{I}_z(x,y,k)$ be a tensor of size $X\times Y\times M/t$ defined by $\widetilde{I}_z(x,y,k)\triangleq I(x,y,z+(k-1)t)$. We define $J^{(z)}(x,y,i)\triangleq\sum_{k=1}^{M/t} P_j^{(k)}(i)\cdot\widetilde{I}_z(x+s_1,y+s_2,k)$ which is a point-wise convolution of $\widetilde{I}_z$. Next, we define $J_i(x,y,z)\triangleq J^{(z)}(x,y,i)$ and let $J$ be the reshaped tensor which is the concatenation of $J_1,\ldots,J_s$, that is $J(x,y,z+(i-1)t)=J_i(x,y,z)$. Now Equation 5 can be rewritten as

$$\widehat{O}(x,y,j) = \sum_{i=1}^{s}\sum_{s_1,s_2}\sum_{z=1}^{t} B_i(s_1,s_2,z)\cdot J(x+s_1,y+s_2,z+(i-1)t). \tag{6}$$

Finally, Equation 6 is just a group-wise convolution of the tensor $J$ with $s$ groups.

Let us compute the computational cost (MAdds) of the implementation for BlkSConv. By Equation 5 (Step a in Figure 2), the computational cost of $s$ point-wise convolutions on the concatenation of $\widetilde{I}_1,\ldots,\widetilde{I}_t$ is $sXYM$. In addition, by Equation 6 (Step b in Figure 2), the computational cost of the group-wise convolution on the tensor $J$ is $sXYt\ell^2$. Finally, the computational cost of the point-wise summation in the last step is $sXY$. The total MAdds of a BlkSConv operation is $sXY(M+t\ell^2+1)$ while the MAdds of a standard convolution is $XYM\ell^2$. Again, let us take convolutions in the last stage of ResNet-34 as examples. We set $s=5$ and $t=4$ as the hyperparameters of the BlkSConv-based convolution. Now the ratio between the MAdds of a BlkSConv-based convolution and a standard convolution is approximately 0.595. Thus the proposed BlkSConv operation is much more efficient than the standard convolution in practical cases. We remark that the proposed implementation requires much GPU memory due to using many group-wise and pointwise convolutions.

## 3.3 Hyperparameter search via principal component analysis

Designing a BlkSConv-based CNN involves hyperparameters including the block depth and the number of basis block vectors in each convolutional layer that affect the performance of the corresponding CNN model. To realize an efficient BlkSConv-based CNN, we conduct a hyperparameter search algorithm based on principal component analysis of trained CNNs. Given a trained CNN, the algorithm generates the block depth and the number of basis block vectors for each standard convolutional layer

of the trained CNN in the following way. First, for each individual $\ell \times \ell \times M$ kernel $K$ of the trained CNN where we assume that $M = 2^\alpha$ for some $\alpha \in \mathbb{N}$, the kernel $K$ is partitioned into $M/t$ block vectors $B_1, B_2, \ldots, B_{M/t}$ each of size $\ell \times \ell \times t$ with $t \in \{1, 2, \ldots, 2^\beta\}$ for some integer $\beta < \alpha$. Next, we perform principal component analysis (PCA) on the set $\{B_1, B_2, \ldots, B_{M/t}\}$. Then, for a fixed integer $\gamma$ and, for each $q \in \{1, 2, \ldots, \gamma\}$, the algorithm computes the variance $V_{t,q}$ of the kernel $K$ which is explained by the first $q$ principal components PC1,PC2,..,PCq. In addition, let $CC_{t,q}$ and $MS_{t,q}$ denote the MAdds and the model size of the BlkSConv under the setting that the block depth is $t$ and the number of basis block vectors is $q$, respectively. Note that the MAdds and the model size of the standard convolution is exactly $CC_{M,1}$ and $MS_{M,1}$, respectively. After computing all $V_{t,q}$, $CC_{t,q}$, and $MS_{t,q}$, the algorithm generates the feasible set

$$H_{\alpha_v, \alpha_c, \alpha_s} \quad = \quad \{(t,q) : V_{t,q} \geq \alpha_v, CC_{t,q} \leq \alpha_c \cdot CC_{M,1}, \text{ and } MS_{t,q} \leq \alpha_s \cdot MS_{M,1}\} \quad (7)$$

for fixed positive constants $\alpha_v, \alpha_c, \alpha_s \in (0,1)$. Finally, the algorithm chooses the hyperparameter $(t,q)$ from $H_{\alpha_v, \alpha_c, \alpha_s}$ according to the computational cost or the model size.

On one hand, note that the goal of BlkSConv is to maintain the prediction performance of the trained standard CNN. In general, the prediction accuracy is proportional to the model size of the constructed CNN. Therefore, in this sense, we choose the hyperparameters $(\hat{t}, \hat{q})$ from $H_{\alpha_v, \alpha_c, \alpha_s}$ such that the constructed BlkSConv has the largest parameter size, that is

$$(\hat{t}, \hat{q}) \quad = \quad \arg \max_{(t,q) \in H_{\alpha_v, \alpha_c, \alpha_s}} MS_{t,q}. \quad (8)$$

One can expect that the generated BlkSConv-based CNN has nice prediction performance compared to the original CNN with standard convolutions.

On the other hand, one of the advantage of BlkSConv operations is that BlkSConv can greatly reduce the model size of the original standard CNN. Thus, in this sense, we can select the hyperparameters $(\tilde{t}, \tilde{q})$ from $H_{\alpha_v, \alpha_c, \alpha_s}$ such that the constructed BlkSConv has the smallest parameter size, that is

$$(\tilde{t}, \tilde{q}) \quad = \quad \arg \min_{(t,q) \in H_{\alpha_v, \alpha_c, \alpha_s}} MS_{t,q}. \quad (9)$$

However, the prediction performance may degrade when the parameter size of the BlkSConv-based model decreases. We will demonstrate in the experimental section that the BlkSConv-based CNNs generated according to Equation 8 also have acceptable prediction accuracy compared to the standard CNNs.

In summary, both Equation 8 and Equation 9 provide ways to determine hyperparameters from the feasible set $H_{\alpha_v, \alpha_c, \alpha_s}$ such that corresponding BlkSConv-based CNNs have smaller model size and fewer multiply-adds operations than the original CNN with standard convolutions.

Finally, let us consider the extreme case that two constants $\beta$ and $\gamma$ are set by $\beta = 0$ and $\gamma = 1$. Let us further set the search parameter $\alpha_v = 0$. Under this restricted search condition, the cardinality of the feasible set $H_{0, \alpha_c, \alpha_s}$ is always 1. Thus the outputs of Equation 8 and Equation 9 are the same. In fact, the resulting BlkSConv-based CNN is exactly the same as the CNN where the standard convolutions are replaced by the blueprint separable convolutions previously developed in (4).

## 4   Experiments

We evaluate BlkSConv and the proposed hyperparameter architecture search algorithm combining with ResNet-10, ResNet-18, and ResNet-26 (5) on ImageNet (22), Stanford Dogs, (14), and Oxford 102 Flowers (19). The proposed methods are also evaluated combining with ResNet-20 and ResNet-56 on CIFAR 10/100 (15).

### 4.1   Hyperparameter Search Details

We apply the PCA-based hyperparameter search algorithm (HSA) developed in Section 3.3 on several variants of ResNet models. In the first part, we consider the large-scale classification scenarios. Several standard ResNets are trained on ImageNet first and their architectures are shown in Table 1. The HSA for searching BlkSConv architectures is only applied to conv3_x, conv4_x, conv5_x layers of these standard ResNets. Next, the search hyperparameters $\alpha_v, \alpha_c, \alpha_s$ are set as 0.5 or

Table 1: ResNet architectures used in the first part of the experiment on ImageNet, Stanford Dogs, and Oxford 102 Flowers. The PCA-based HSA is applied to conv3_x, conv4_x, conv5_x layers and the corresponding convolutional kernel is replaced by the BlkSConv module found by HSA.

| ResNet-10 (L=1), ResNet-18 (L=2), ResNet-26 (L=3) | | | | |
|---|---|---|---|---|
| Layers Names | Output Size | ResNet | Applying HSA | e.g.(ResNet-10) |
| conv1
max pool | $112 \times 112 \times 64$
$56 \times 56 \times 64$ | $7 \times 7$, 64, stride 2
$3 \times 3$, stride 2 | No | |
| conv2_x | $56 \times 56 \times 64$ | $\begin{bmatrix} 3 \times 3, & 64 \\ 3 \times 3, & 64 \end{bmatrix} \times L$ | No | |
| conv3_x | $28 \times 28 \times 128$ | $\begin{bmatrix} 3 \times 3, & 128 \\ 3 \times 3, & 128 \end{bmatrix} \times L$ | Yes | conv-s5t2 |
| conv4_x | $14 \times 14 \times 256$ | $\begin{bmatrix} 3 \times 3, & 256 \\ 3 \times 3, & 256 \end{bmatrix} \times L$ | Yes | conv-s5t2 |
| conv5_x | $7 \times 7 \times 512$ | $\begin{bmatrix} 3 \times 3, & 512 \\ 3 \times 3, & 512 \end{bmatrix} \times L$ | Yes | conv-s1t1 |
| average pool
fully connected | $1 \times 1 \times 512$
1000 | $7 \times 7$
$512 \times 1000$ fc | | |

Table 2: Performance results for BlkSConv-based ResNet-18 and ResNet-26 on ImageNet.

| $(\alpha_v, \alpha_c, \alpha_s, SS)$ | ResNet-18 on ImageNet | | | ResNet-26 on ImageNet | | |
|---|---|---|---|---|---|---|
| | Accuracy | P_ratio | MA_ratio | Accuracy | P_ratio | MA_ratio |
| $(0.50, 0.50, 0.50, \max)$ | 69.922 | 0.4065 | 0.4614 | 72.038 | 0.4241 | 0.4618 |
| $(0.50, 0.75, 0.75, \max)$ | 69.782 | 0.6014 | 0.6597 | 72.326 | 0.6308 | 0.6722 |
| $(0.50, 0.50, 0.50, \min)$ | 67.572 | 0.1264 | 0.2700 | 69.970 | 0.1250 | 0.2668 |
| $(0.50, 0.75, 0.75, \min)$ | 67.540 | 0.1246 | 0.2986 | 69.922 | 0.1243 | 0.2910 |
| Standard (replaced layers) | 70.728 | 10.8M | 1213.8M | 72.604 | 17.03M | 1907.4M |

0.75. It is possible that the feasible set $H_{\alpha_v, \alpha_c, \alpha_s}$ is empty. In this case, the corresponding standard convolutional layer is not replaced and is denoted by conv as shown in Table 1. Moreover, the proposed HSA has two selection strategies: one is based on the largest parameter size, denoted by $SS = \max$, and the other is based on the smallest parameter size, denoted by $SS = \min$ as shown in Table 2. The selected BlkSConv, $sitj$, which means $i$ basis block vectors and $j$ depth of the blocks.

The feasible set $H_{\alpha_v, \alpha_c, \alpha_s}$ is likely to be empty when the parameter $\alpha_v$ is large. In the case that $\alpha_v$ is large, it often requires many principal components to accumulate enough explained variance and thus this causes large numbers of parameters or MAdds. Therefore, the feasible set $H_{\alpha_v, \alpha_c, \alpha_s}$ is probably empty when we further require small $\alpha_c$ and $\alpha_s$. On the other hand, the parameter $\alpha_v$ cannot be too small because the prediction performance of the network is highly proportional to the amount of the accumulated variance as discussed in Section 3.3 where we will demonstrate it in the ablation study of this section. For the above reason, we only present the results for ResNet-18 and ResNet-26 on ImageNet under the setting that $\alpha_v = 0.5$ which are shown in Table 2.

In the second part, we consider the small-scale classification on CIFAR10/100. We use the standard ResNet-20 and ResNet-56 as the experimental models where the architectures are slightly modified to suit the small-scale images. The proposed HSA is only applied to conv4_x layers of these two standard ResNets. More BlkSConv-based architecture search results can be found in the appendix.

## 4.2 Performance on large-scale classification: ImageNet

To evaluate the performance of BlkSConv-based models in large-scale recognition, we conduct experiments on ImageNet(22). Each model takes 3 days to be trained on a single GPU (Nvidia Tesla V100). ImageNet contains nearly 1.3M training images and 50,000 testing images. For

Table 3: Comparison among the BlkSConv-based and Standard ResNet on ImageNet and CIFAR.

| Dataset | Models | Accuracy | Parameters | MAdds |
|---|---|---|---|---|
| ImageNet | ResNet-10 standard | 63.386 | 4.64M | 520M |
| | BlkSConv-ResNet-18 $(0.5, 0.5, 0.5, \max)$ | 69.922 | 4.39M | 560M |
| | BlkSConv-ResNet-26 $(0.5, 0.5, 0.5, \min)$ | 69.970 | 2.13M | 509M |
| CIFAR 100 | ResNet-20 standard | 67.994 | 202752 | 12.97M |
| | BlkSConv-ResNet-20 $(0.5, 0.5, 0.5, \max)$ | 67.078 | 72704 | 5.04M |
| | BlkSConv-ResNet-56 $(0.5, 0.5, 0.5, \min)$ | 69.994 | 149440 | 10.61M |

Table 4: Performance results for BlkSConv-based ResNet-56 on CIFAR10/100.

| $(\alpha_v, \alpha_c, \alpha_s, SS)$ | ResNet-56 on CIFAR 10 | | | ResNet-56 on CIFAR 100 | | |
|---|---|---|---|---|---|---|
| | Accuracy | P_ratio | MA_ratio | Accuracy | P_ratio | MA_ratio |
| $(0.5, 0.5, 0.5, \max)$ | 93.372 | 0.3734 | 0.3829 | 70.636 | 0.3734 | 0.3829 |
| $(0.5, 0.75, 0.75, \max)$ | 93.338 | 0.6196 | 0.6292 | 70.668 | 0.6196 | 0.6292 |
| $(0.5, 0.5, 0.5, \min)$ | 93.324 | 0.2335 | 0.2462 | 69.994 | 0.2316 | 0.2570 |
| $(0.5, 0.75, 0.75, \min)$ | 93.324 | 0.2335 | 0.2462 | 69.994 | 0.2316 | 0.2570 |
| Standard (replaced layers) | 93.218 | 645120 | 41.28M | 70.998 | 645120 | 41.28M |

the experimental setup, ResNet-10, ResNet-18, and ResNet-26 are trained on ImageNet under the following setting. The number of epochs is 100 and the batch size is 256. SGD is used as the optimizer and the initial learning rate, the momentum, and the weight decay are set to 0.1, 0.9, and $10^{-4}$, respectively. The learning rate is scheduled to decay by a factor of 0.1 at epochs 30, 60, and 90. We augment the data via random resized crop to 224px and random horizontal flip. The performance results are shown in Table 2, More experimental results can be found in the appendix.

On one hand, let us focus the cases that $\alpha_v = 0.5$ and $SS = \max$ in Table 2. The prediction accuracies of the selected BlkSConv-based models and the standard model are close within 1%. It confirms our expectation that BlkSConv-based models have smaller parameter sizes and fewer MAdds than standard models while preserving prediction performance if the proposed HSA adopts a selection strategy based on the maximum parameter size.

On the other hand, let us consider the case that $\alpha_v = 0.5$ and $SS = \min$ in Table 2. The parameters and MAdds of the BlkSConv-based models are only 12.6% and 29.8% of the standard model while the gap of their prediction accuracies is about 3%. We adopt an interesting way based on restricting the parameter size and MAdds to interpret the advantage of the generated BlkSConv-based models where the selection strategy $SS$ is set as min. We also compare the standard ResNet-10, the BlkSConv-based ResNet-18, and the BlkSConv-based ResNet-26 in Table 3 where the parameter sizes or MAdds of three given models are similar. The BlkSConv-based ResNet-26 with parameter $(0.5, 0.5, 0.5, \min)$ and the BlkSConv-based ResNet-18 with parameter $(0.5, 0.5, 0.5, \max)$ greatly outperform the standard ResNet-10 where both the BlkSConv-based models lead to an accuracy gain of at least 6.5%. In addition, the BlkSConv-based ResNet-26 with parameter $(0.5, 0.5, 0.5, \min)$ only has half the parameter size of the BlkSConv-based ResNet-18 with parameter $(0.5, 0.5, 0.5, \max)$.

## 4.3 Performance on small-scale classification: CIFAR 10/100

The performance results are shown in Table 4. The BlkSConv-based ResNet-56 models have much smaller model sizes and fewer MAdds than the standard model while all BlkSConv-based variants outperform the standard model on CIFAR 10 and have comparable accuracies on CIFAR 100. In the bottom of Table 3, the BlkSConv-based ResNet-20 with $(0.5, 0.5, 0.5, \max)$ and the standard ResNet-20 both have a comparable accuracy while the BlkSConv-based ResNet-20 model is compressed 64% of the parameter size and MAdds is decreased 61% compared to the standard ResNet-20 model. Furthermore, the BlkSConv-based ResNet-56 with $(0.5, 0.5, 0.5, \min)$ and the standard ResNet-20 model both have similar parameter sizes and MAdds while the BlkSConv-based ResNet-56 model has an accuracy gain of 2%. More experimental results can be found in the appendix.

Table 5: Performance comparison among BlkSConv-based and the standard ResNet-18 models.

| $(\alpha_v, \alpha_c, \alpha_s, SS)$ | Stanford Dogs | | | Oxford 102 Flowers | | |
|---|---|---|---|---|---|---|
| | Accuracy | P_ratio | MA_ratio | Accuracy | P_ratio | MA_ratio |
| $(0.5, 0.5, 0.5, \max)$ | 53.005 | 0.5327 | 0.5394 | 65.546 | 0.5327 | 0.5394 |
| $(0.5, 0.75, 0.75, \max)$ | 53.359 | 0.7277 | 0.7377 | 64.567 | 0.7277 | 0.7377 |
| $(0.5, 0.5, 0.5, \min)$ | 53.159 | 0.3273 | 0.4006 | 65.289 | 0.4835 | 0.5179 |
| $(0.5, 0.75, 0.75, \min)$ | 53.615 | 0.3171 | 0.4611 | 63.217 | 0.4743 | 0.5872 |
| Standard (replaced layers) | 52.436 | 10.8M | 1213.8M | 62.238 | 10.8M | 1213.8M |

Table 6: Results on Stanford Dogs for different $\alpha_v$ with $SS = \min$.

| ResNet-18 on Stanford Dogs $(\alpha_v, \alpha_c = 0.75, \alpha_s = 0.75, SS = \min)$ | | | | | | | |
|---|---|---|---|---|---|---|---|
| $\alpha_v$ | Accuracy | P_ratio | MA_ratio | $\alpha_v$ | Accuracy | P_ratio | MA_ratio |
| 0.0 | 50.918 | 0.0397 | 0.1781 | 0.3 | 52.839 | 0.1074 | 0.2377 |
| 0.1 | 50.499 | 0.0449 | 0.1777 | 0.4 | 52.035 | 0.2751 | 0.4684 |
| 0.2 | 51.040 | 0.0767 | 0.2458 | 0.5 | 53.615 | 0.3171 | 0.4611 |
| | | | | Standard | 52.436 | 10.8M | 1213.8M |

## 4.4 Performance on fine-grained classification

We conduct experiments for fine-grained recognition on two datasets Stanford Dogs and Oxford 102 Flowers. For the experimental setup, the standard ResNet-18 and its BlkSConv-variants are all trained from scratch by augmenting data through random crops, horizontal flips, and random gamma transform. We use SGD as the optimizer and the initial learning rate, the moment, and the weight decay are set to 0.1, 0.9, and $10^{-4}$, respectively. The number of epochs is 200, and the learning rate is scheduled to decay at epochs 100, 150, and 200 by a factor of 0.1. The proposed BlkSConv-based ResNet-18 models significantly outperform the standard ResNet-18 model both on Stanford Dogs and Oxford 102 Flowers as shown in shown Table 5..

## 4.5 Ablation Study: Necessity to have large explained variance

Here, we demonstrate how the variance hyperparameter $\alpha_v$ affects the prediction accuracy of BlkSConv-based CNNs. We use ResNet-18 as the experimental model. After training the standard ResNet-18 on Stanford Dogs, the next goal is to find several BlkSConv-variants of ResNet-18 all of which have different explained variances such that their accuracies can be compared. Note that $H_{a,0.75,0.75} \subseteq H_{b,0.75,0.75}$ for any $a, b$ with $a \geq b$. Based on this observation, the model in $H_{b,0.75,0.75}$ which has the smallest parameter size is likely to have a small explained variance as well. Therefore, the selection strategy of the proposed HSA is set by $SS = \min$ in order to select several BlkSConv-based ResNet-18 models with different explained variances. Now we apply the proposed HSA to the trained ResNet-18 under six search hyperparameters $\{(\alpha_v, 0.75, 0.75, \min) : \alpha_v = 0.0, 0.1, 0.2, \ldots, 0.5\}$. The comparison result is shown in Table 6. It can be seen that the accuracy of the BlkSConv-based model is greater than that of the standard model only when the variance hyperparameter $\alpha_v$ is large enough, that is $\alpha_v \geq 0.5$.

## 5 Conclusion

In this paper, we introduce the block-wise separable convolutions (BlkSConv) to replace standard convolutions. An efficient implementation of the BlkSConv operation via a combination of pointwise and group-wise convolutions is also given. Moreover, we also propose an efficient hyperparameter search algorithm based on principal component analysis in order to select an optimal BlkSConv-based convolutional network under certain constraints on model size and model efficiency. Finally, the experimental results demonstrate the advantage of the BlkSConv-based CNN models selected by the proposed hyperparameter search algorithm.

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
