# OpenReview forum: "Block-wise Separable Convolutions: An Alternative Way to Factorize Standard Convolutions"
_NeurIPS.cc/2022/Conference — NeurIPS 2022 Submitted_

### Official Review · Reviewer_B29N · 2022-07-12

**Rating:** 4
**Confidence:** 3
**Soundness:** 3 good
**Presentation:** 3 good
**Contribution:** 2 fair

**Summary:**

This paper introduces a novel convolution factorization for efficient image recognition. The formalization proceeds as follows: First, the authors rewrite the standard convolution (on a single output channel) as the sum of contributions of the application of different kernels to different groups of input channels (t groups). Then, every kernel for every group is formalized as a low-rank approximation achieved by means of a linear combination of s basis kernels. As this approach effectively reduces the number of trainable parameters but does not affect (or even increases) the number of operations, the authors showcase an efficient implementation. Specifically, exploiting the linearity of all parts involved (both the convolution and the decomposition of the kernel into basis vectors), they show the factorized convolution can be espressed in terms of standard operations such as point-wise convolution, reshaping, and group convolutions. Finally, the authors propose a PCA-based technique to automatically search for suitable values of t and s, that enable a good tradeoff between accuracy and computation. Experiments are carried out with ResNets variants on 5 datasets.

**Questions:**

N/A

**Limitations:**

- The authors briefly mention memory overhead as a potential limitation. I encourage them to discuss about on time latency measurements as well, in the case the proposed model is not beneficial in that respect.
- No potential negative societal impact that I can think of.

**Strengths And Weaknesses:**

PROs

- The proposed low-rank formulation of convolutional kernels seems novel to me. Moreover, the authors show how it is easily implementable as a combination of standard neural network operations.
- The paper is clear for most parts, and the authors succeed in explaining all the operations involved in a clear way. To the best of my judgment, the technical contribution is sound and correct.
- The experimental section showcases good performances in all the tested settings, with the proposed convolution factorization being able to reach performance within 1-3 points of accuracy of the original model (or even surpassing its performance) at a fraction of the computational cost. Remarkably, Table 5 illustrates how optimizing a ResNet with the proposed factorization is a better strategy that using a smaller version of the architecture (e.g. BlkSConv-ResNet-18 or BlkSConv-ResNet-26 are better than ResNet-10 on ImageNet).
- The authors release their code as part of the supplementary material, which is considerable.


CONs

- The main issue with the paper is that in the experimental section it does not provide any baseline for the proposed factorization. It is true that the block-wise separable convolution provides a much better accuracy/efficiency tradeoff with respect to standard convolutions. However, several other methods have been popularized in the past to save computations. For instance, how would group convolutions, depthwise separable convolutions, low-rank factorization methods [4,5] or channel pruning [6] perform in the same settings the authors use? Without reporting at least a subset of them as baselines, it is very hard to quantify whether the proposed factorization, despite being beneficial, would be something interesting and impactful for the community
- The paper misses the discussion of model compression prior works that rely on low-rank factorization of weight matrices. For instance, [4,5] are seminal approaches that use SVD decomposition to factorize convolutional kernels as a product of lower rank matrices. Since here a low-rank factorization is also achieved, I think a discussion of those models would improve the paper's quality.
- Formalizing a convolutional kernel as a linear combination of a set of basis has recently raised a lot of interest, especially in the context of conditional computation. For instance, models such as [1,2,3] predict the combination coefficients based on the input feature itself. Although this is somewhat different with what is proposed here, I think the paper misses a discussion of this branch of related works, for completeness.
- The proposed implementation described in Sec 3.2 incurs into a higher memory overhead	and involves some heavy memory-bound operations such as reshaping and copying. I therefore expect that, despite the reduction in MAdds, the actual on device latency of the proposed convolution is similar to (or even higher than) the original one. I think this should be verified and discussed in the manuscript.
- Fig 1 is helpful but could be made more clear:
  - Fig. 1 left should specify, for clarity, that it represents the special case for s=1.
  - The color coding (green) suggests that the kernel is shared among the N output channels. To my understanding, that's not the case.

Typos:
- line 138/148: "top of Figure 1" / "bottom of Figure 1" -> "Figure 1 left" / "Figure 1 right"

In summary, I think this is a good paper with the potential of being impactful, yet it needs to report some baseline performances to put the results into perspective.

[1] Chen, Yinpeng, et al. "Dynamic convolution: Attention over convolution kernels." Proceedings of the IEEE/CVF Conference on Computer Vision and Pattern Recognition. 2020.

[2] Li, Yunsheng, et al. "Revisiting dynamic convolution via matrix decomposition." ICLR. 2021.

[3] Zhang, Mingda, et al. "Basisnet: Two-stage model synthesis for efficient inference." Proceedings of the IEEE/CVF Conference on Computer Vision and Pattern Recognition. 2021.

[4] Zhang, Xiangyu, et al. "Accelerating very deep convolutional networks for classification and detection." IEEE transactions on pattern analysis and machine intelligence 38.10 (2015): 1943-1955.

[5] Jaderberg, Max, Andrea Vedaldi, and Andrew Zisserman. "Speeding up convolutional neural networks with low rank expansions." arXiv preprint arXiv:1405.3866 (2014).

[6] He, Yihui, Xiangyu Zhang, and Jian Sun. "Channel pruning for accelerating very deep neural networks." Proceedings of the IEEE international conference on computer vision. 2017.

---

> ### Author Response · Authors · 2022-08-02
> **Response to Reviewer B29N**
>
> 1. The main issue with the paper is that in the experimental section it does not provide any baseline for the proposed factorization. It is true that the block-wise separable convolution provides a much better accuracy/efficiency tradeoff with respect to standard convolutions. However, several other methods have been popularized in the past to save computations. For instance, how would group convolutions, depthwise separable convolutions, low-rank factorization methods [4,5] or channel pruning [6] perform in the same settings the authors use? Without reporting at least a subset of them as baselines, it is very hard to quantify whether the proposed factorization, despite being beneficial, would be something interesting and impactful for the community.
>
> Thanks for the reviewer’s comment. We add experimental results of MobileNetV1, MobileNetV2, and MobileNetV3 as baselines for comparing with proposed BlkSConv-based ResNet models. We also conduct the latency experiments on GPU and CPU on these models. We record the average of 300 running times on these models. The results are shown in J7xD 1) response.
>
> 2. The paper misses the discussion of model compression prior works that rely on low-rank factorization of weight matrices. For instance, [4,5] are seminal approaches that use SVD decomposition to factorize convolutional kernels as a product of lower rank matrices. Since here a low-rank factorization is also achieved, I think a discussion of those models would improve the paper's quality.
>
> Thanks for the reviewer’s valuable comment. In general, previous models based on low-rank factorization are carried out in the following way. In general, a few standard convolutional operations are applied to the input tensor first and then a certain number (equals the channel size of the output tensor) point-wise convolutions are applied to the generated tensor.  To contrast the proposed BlkSConv with the low-rank-factorization-based convolutions, several point-wise convolutions are applied to the input tensor first and then group-wise convolution is applied to the generated tensor. That is the main difference between these two approaches.
>
> Thanks to the reviewer’s suggestion, we would like to add a discussion of those models in the modified paper.
>
> 3. Formalizing a convolutional kernel as a linear combination of a set of basis has recently raised a lot of interest, especially in the context of conditional computation. For instance, models such as [1,2,3] predict the combination coefficients based on the input feature itself. Although this is somewhat different with what is proposed here, I think the paper misses a discussion of this branch of related works, for completeness.
>
> To our best knowledge, dynamic convolutions are mainly constructed based on the attention mechanism. It is known that the main component of an attention operator is a linear transformation between the query, key, and context vectors which is somewhat similar to our approach. Thanks to the reviewer’s suggestion, we would add a discussion in the related works of dynamic convolutions.
>
> 4. The proposed implementation described in Sec 3.2 incurs into a higher memory overhead and involves some heavy memory-bound operations such as reshaping and copying. I therefore expect that, despite the reduction in MAdds, the actual on device latency of the proposed convolution is similar to (or even higher than) the original one. I think this should be verified and discussed in the manuscript.
>
> Thanks for the reviewer’s valuable comments. As the reviewer’s expectation, the actual latency of the BlkSConv is higher than that of the standard convolution while the proposed BlkSConv has smaller MAdds than the standard convolution. This fact is verified in the table shown in the J7xD 1) response. We would like to discuss this fact in our modified manuscript. Thanks to the reviewer again for the great comment.
>
> 5. Fig 1 is helpful but could be made more clear: (a)Fig. 1 left should specify, for clarity, that it represents the special case for s=1. (b) The color coding (green) suggests that the kernel is shared among the N output channels. To my understanding, that's not the case.
>
>  Thanks for the reviewer’s comment. We modified Fig.1 to make it consistent with the content of the proposed paper.
>
> 6. Typos: line 138/148: "top of Figure 1" / "bottom of Figure 1" -> "Figure 1 left" / "Figure 1 right"
>
> Thanks for the reviewer’s comment. We corrected that typo.

---

### Official Review · Reviewer_J7xD · 2022-07-12

**Rating:** 5
**Confidence:** 4
**Soundness:** 3 good
**Presentation:** 3 good
**Contribution:** 3 good

**Summary:**

This paper proposes block-wise factorization for approximating convolutions. Given a k x k x M convolutional kernel, it factorizes it into many smaller blocks k x k x t, and then use PCA to select a subset of these blocks to approximate the original more expensive computation.  Overall, kernel factorization is a somewhat common way to reduce ConvNet computations (previous  depthwise/pointwise/group convs are more common), but the new PCA based block selection is kind of new and interesting. Results on CIFAR and ImageNet show the proposed approach can achieve better accuracy-computation trade-offs with various ResNet networks.

**Questions:**

see above.

**Ethics Review Area:**

["I don’t know"]

**Limitations:**

No concern on social impact.

**Strengths And Weaknesses:**

==== Strengths ====
1. Interesting idea on using PCA to choose the most important top-k sub blocks. To my knowledge, this seems to be quite new and different from previous factorization.
2. Well written paper with promising results.

==== Weaknesses ====
1. No latency is reported: Table 3 and 4 show impressive reduction in parameters and MAdds, but it is unclear how fast are these models in real hardwares. Would be nice to include latency measurements.
2. Limited experiments: most experiments are based on ResNets, would be better to explore other models  (especially those more FLOPs-efficient models such as EfficientNets or MobileNetV3).
3. Some details are not very clear: for example, it requires both M and t to be the form of 2^k, but what if the original network has irregular channel sizes (such as 40)?

---

> ### Author Response · Authors · 2022-08-02
> **Response to Reviewer J7xD**
>
> 1. No latency is reported: Table 3 and 4 show impressive reduction in parameters and MAdds, but it is unclear how fast are these models in real hardwares. Would be nice to include latency measurements.
>
> Thanks for the reviewer’s comment. We add experimental results of MobileNetV1, MobileNetV2, and MobileNetV3 as baselines for comparing with proposed BlkSConv-based ResNet models. Due to limited time, we adopt the experimental results from the work of Haase and Amthor proposed in CVPR 2020 since our training setting is the same as theirs. The results are shown as follows.
>
> We conduct the latency experiments on GPU and CPU on these models. We record the average of 300 running times on these models.
>
> - GPU: V100-32GB
> - CPU: Intel(R) Xeon(R) Gold 6248 CPU @ 2.50GHz
>
> | Model | Acc | Params | MAdds | latency-gpu | latency-cpu |
> | :----- | :--- | ------: | -----: | ----------------: | ----------------: |
> | MobileNetV1 (x0.25) | 51.8 [1] | 0.47M | 46.68M | 3.56 ms | 8.77 ms |
> | MobileNetV1 (x0.5) | 63.5 [1] | 1.33M | 160.21M | 3.87 ms | 11.91 ms |
> | MobileNetV1 (x0.75) | 68.2 [1] | 2.58M | 341.16M | 3.50 ms | 13.84 ms |
> | MobileNetV1 (x1.0) | 70.8 [1] | 4.23M | 589.56M | 3.88 ms | 15.98 ms |
> | MobileNetV2 (x1.0) | 69.7 [1] | 3.50M | 342.60M | 8.09 ms | 26.83 ms |
> | MobileNetV3-small (x1.0) | 64.4 [1] | 1.66M | 66.24M | 8.40 ms | 18.34 ms |
> | MobileNetV3-large (x1.0) | 71.5 [1] | 3.93M | 237.90M | 9.38 ms | 25.94 ms |
> | BlkSConv-ResNet18 (0.5,0.5,0.5,max) | 69.9 | 4.39M | 560M | 4.51 ms | 15.47 ms |
> | BlkSConv-ResNet18 (0.5,0.5,0.5,min) | 67.6 | 1.36M | 323.72M | 4.31 ms | 17.24 ms |
> | BlkSConv-ResNet26 (0.5,0.5,0.5,min) | 70.0 | 2.13M | 509M | 6.44 ms | 24.49 ms |
> | Standard ResNet10 | 63.4 | 4.64M | 520M | 1.91 ms | 8.94 ms |
> | Standard ResNet18 | 70.8 | 10.8M | 1213.8M | 3.33 ms | 14.98 ms |
> | Standard ResNet26 | 72.6 | 17.03M | 1907.4M | 4.60 ms | 21.72 ms |
>
> For these listed models, we provide not only their accuracy performance but also their latency times both in GPU and CPU.
>
> First of all, we compare BlkSConv-ResNet26 with MobileNetV1(x1.0) where these two models have similar accuracy performance. According to the above table, BlkSConv-ResNet26 has smaller size and MAdds than MobileNetV1(x1.0) in a theoretical sense while MobileNetV1(x1.0) has smaller latency time than BlkSConv-ResNet26 in a real sense.
>
> Moreover, we compare BlkSConv-ResNet26 with MobileNetV2(x1.0) and MobileNetV3-large(x1.0) where these three models also have similar accuracy performance.  According to the above table, BlkSConv-ResNet26 has larger size and MAdds than MobileNetV2(x1.0) and MobileNetV3-large(x1.0) while BlkSConv-ResNet26 has smaller latency time than MobileNetV2(x1.0) and MobileNetV3-large(x1.0).
>
> We will add these comparison results in our modified paper.
>
> 2. Limited experiments: most experiments are based on ResNets, would be better to explore other models (especially those more FLOPs-efficient models such as EfficientNets or MobileNetV3).
>
> The response is the same as the discussion in J7xD 1) answer.
>
> 3. Some details are not very clear: for example, it requires both M and t to be the form of 2^k, but what if the original network has irregular channel sizes (such as 40)?
>
> Thanks for the reviewer’s comment. In our proposed model, it is only required that M is divisible by t. Since the models used in our experiment are ResNets where the channel sizes of the generated tensors are all in the form of 2^k (that is M=2^k), we required that t is of the form of 2^p to satisfy that M can be divided by t in these ResNet models.  Therefore, we do not require that both M and t to be the form of 2^k for the general models.
>
> [1] D. Haase and M. Amthor, "Rethinking Depthwise Separable Convolutions: How Intra-Kernel Correlations Lead to Improved MobileNets," in Proceedings of the IEEE Conference on Computer Vision and Pattern Recognition (CVPR), pages 14600–14609, 2020.

---

> > ### Comment · Reviewer_J7xD · 2022-08-08
> > **Not convinced by the new results**
> >
> > Thanks for your response, and I particularly appreciate the new comparison to MobileNets.
> >
> > However, the performance of MobileNet shown on this table doesn't match the original paper. According to the original paper (Searching for MobileNetV3, https://arxiv.org/pdf/1905.02244.pdf), MobileNetV3-Small (x1.0) has top-1 accuracy 67.4%, instead of 64.45% reported here.  In other words, MobileNetV3-small has similar accuracy as BlkSConv-ResNet18, but has much less FLOPs.
> >
> > The latency results are also interesting:  BlkSConv-ResNet18 can reduce the params/flops, but has much larger latency and lower accuracy than the original ResNet18. In this case, I am not convinced if the proposed method is practically useful.
> >
> > Regarding the last question: M has to be 2^k, but many models have channel sizes not in the form of 2^k (such as MobileNetV3, EfficientNet, RegNet, ResNeSt, FBNet). Does it mean you won't be able to optimize these models with the current method?

---

> > > ### Author Response · Authors · 2022-08-09
> > > **Latency and hardware optimization**
> > >
> > > 1. The latency results are also interesting: BlkSConv-ResNet18 can reduce the params/flops, but has much larger latency and lower accuracy than the original ResNet18.
> > >
> > > - The latency results shown in the table depend on the hardware optimization. In particular, in the papers [2,3], it is shown that the flops are not positively related to the latency because of the hardware optimization of the specific operations. So using different devices might result in different latency.
> > >
> > > [2] Dai, Xiaoliang, et al. "Chamnet: Towards efficient network design through platform-aware model adaptation." Proceedings of the IEEE/CVF Conference on Computer Vision and Pattern Recognition. 2019.
> > >
> > > [3] Xiong, Yunyang, et al. "Mobiledets: Searching for object detection architectures for mobile accelerators." Proceedings of the IEEE/CVF Conference on Computer Vision and Pattern Recognition. 2021.
> > >
> > > 2. Regarding the last question: M has to be 2^k, but many models have channel sizes not in the form of 2^k (such as MobileNetV3, EfficientNet, RegNet, ResNeSt, FBNet). Does it mean you won't be able to optimize these models with the current method?
> > >
> > > - As the previous response, M is not necessary to be 2^k. The requirement in our proposed method is that M is divisible by t.
> > >
> > > - Our method can optimize convolutional models if they have standard convolutional blocks.

---

> > > > ### Comment · Reviewer_J7xD · 2022-08-09
> > > > **Borderline**
> > > >
> > > > Thanks for clarifying that M is not necessary to be 2^k!
> > > > I agree that FLOPs is not necessary correlated to latency, but if the proposed method makes the original ResNet18 slower (and also lower accuracy), then the contribution of reducing FLOPs/Prams is limited in practice.

---

### Official Review · Reviewer_tB5v · 2022-07-12

**Rating:** 4
**Confidence:** 4
**Soundness:** 3 good
**Presentation:** 2 fair
**Contribution:** 3 good

**Summary:**

This works focuses on CNN-based model compression. It first views the kernel in the standard convolution as an ordered set of vectors, where each vector has smaller dimension than the inputs. Each vector can then be treated as a linear combination of another group of basis vectors. By limiting the number of the basis vectors, it approximates the standard convolutional kernel. The proposed operator is called block-wise separable convolutions (BlkSConv) and can be implemented as a combination of point-wise and group-wise convolutions. The paper also proposes a PCA-based method to determine the dimension and the number of basis vectors. By replacing some standard convolutions with the proposed BlkSConv in CNNs, the authors demonstrate the effectiveness of BlkSConv in CNN-based model compression.

**Questions:**

Does the fact that BlkSConv can be implemented as a combination of point-wise and group-wise convolutions indicate some insights on how BlkSConv works from another point of view?

**Strengths And Weaknesses:**

Strengths:
The paper provides a unique view of convolutional kernels and derives the proposed BlkSConv. Meanwhile, the authors find an easy way to implement BlkSConv. It also proposes a systematic way to determine hyperparameters in BlkSConv. All of these make the proposed method easy and practical to use.

Weaknesses:
1. The notations in figures do not match those in texts. It makes the paper less clear.
2. The experiments only compare with regular ResNet. Comparisons with other model compression works are necessary.

---

> ### Author Response · Authors · 2022-08-02
> **Response to Reviewer tB5v**
>
> 1. The notations in figures do not match those in texts. It makes the paper less clear.
>
> Thanks for the reviewer’s comment. We modified Fig.1 to make it consistent with the content of the proposed paper.
>
> 2. The experiments only compare with regular ResNet. Comparisons with other model compression works are necessary.
>
> Thanks for the reviewer’s comment. We add experimental results of MobileNetV1, MobileNetV2, and MobileNetV3 as baselines for comparing with proposed BlkSConv-based ResNet models. Due to limited time, we adopt the experimental results from the work of Haase and Amthor proposed in CVPR 2020 since our training setting is the same as theirs. The results are shown as follows.
>
> We conduct the latency experiments on GPU and CPU on these models. We record the average of 300 running times on these models.
>
> - GPU: V100-32GB
> - CPU: Intel(R) Xeon(R) Gold 6248 CPU @ 2.50GHz
>
> | Model | Acc | Params | MAdds | latency-gpu | latency-cpu |
> | :----- | :--- | ------: | -----: | ----------------: | ----------------: |
> | MobileNetV1 (x0.25) | 51.8 [1] | 0.47M | 46.68M | 3.56 ms | 8.77 ms |
> | MobileNetV1 (x0.5) | 63.5 [1] | 1.33M | 160.21M | 3.87 ms | 11.91 ms |
> | MobileNetV1 (x0.75) | 68.2 [1] | 2.58M | 341.16M | 3.50 ms | 13.84 ms |
> | MobileNetV1 (x1.0) | 70.8 [1] | 4.23M | 589.56M | 3.88 ms | 15.98 ms |
> | MobileNetV2 (x1.0) | 69.7 [1] | 3.50M | 342.60M | 8.09 ms | 26.83 ms |
> | MobileNetV3-small (x1.0) | 64.4 [1] | 1.66M | 66.24M | 8.40 ms | 18.34 ms |
> | MobileNetV3-large (x1.0) | 71.5 [1] | 3.93M | 237.90M | 9.38 ms | 25.94 ms |
> | BlkSConv-ResNet18 (0.5,0.5,0.5,max) | 69.9 | 4.39M | 560M | 4.51 ms | 15.47 ms |
> | BlkSConv-ResNet18 (0.5,0.5,0.5,min) | 67.6 | 1.36M | 323.72M | 4.31 ms | 17.24 ms |
> | BlkSConv-ResNet26 (0.5,0.5,0.5,min) | 70.0 | 2.13M | 509M | 6.44 ms | 24.49 ms |
> | Standard ResNet10 | 63.4 | 4.64M | 520M | 1.91 ms | 8.94 ms |
> | Standard ResNet18 | 70.8 | 10.8M | 1213.8M | 3.33 ms | 14.98 ms |
> | Standard ResNet26 | 72.6 | 17.03M | 1907.4M | 4.60 ms | 21.72 ms |
>
> For these listed models, we provide not only their accuracy performance but also their latency times both in GPU and CPU.
>
> First of all, we compare BlkSConv-ResNet26 with MobileNetV1(x1.0) where these two models have similar accuracy performance. According to the above table, BlkSConv-ResNet26 has smaller size and MAdds than MobileNetV1(x1.0) in a theoretical sense while MobileNetV1(x1.0) has smaller latency time than BlkSConv-ResNet26 in a real sense.
>
> Moreover, we compare BlkSConv-ResNet26 with MobileNetV2(x1.0) and MobileNetV3-large(x1.0) where these three models also have similar accuracy performance.  According to the above table, BlkSConv-ResNet26 has larger size and MAdds than MobileNetV2(x1.0) and MobileNetV3-large(x1.0) while BlkSConv-ResNet26 has smaller latency time than MobileNetV2(x1.0) and MobileNetV3-large(x1.0).
>
> We will add these comparison results in our modified paper.
>
> 3. Questions: Does the fact that BlkSConv can be implemented as a combination of point-wise and group-wise convolutions indicate some insights on how BlkSConv works from another point of view?
>
> BlkSConv is originally developed in the sense that the standard convolutional kernel can be approximated by a linear combination of few basis block vectors. It is also shown in this paper that the BlkSConv operator can be implemented as a combination of point-wise and group-wise convolutions. We believe as the reviewer suggested that there may exist a better explanation for the BlkSConv operator. Currently, we are not able to indicate what it is. Nevertheless, we will try our best to understand what it could be in our future work. Thanks to the reviewer again for suggesting us this topic.
>
> [1] D. Haase and M. Amthor, "Rethinking Depthwise Separable Convolutions: How Intra-Kernel Correlations Lead to Improved MobileNets," in Proceedings of the IEEE Conference on Computer Vision and Pattern Recognition (CVPR), pages 14600–14609, 2020.

---

### Meta-Review · Area_Chair_BZ5N · 2022-08-24

**Recommendation:** Reject
**Confidence:** Certain

**Metareview:**

This paper proposes a factorized convolution operation, which can be implemented with a combination of group wise convolution and point wise convolution. The reviewers in general agreed that this a valid idea, and found the PCA based hyper-parameter search method interesting. However, several concerns were raised on the experimental side, including lack of fair comparison with other efficient convolution baselines, as well as degraded latency and memory cost compared to simple ResNets. Given the practical nature of the work, the AC agrees that this work needs substantial improvements on the empirical evaluations to pass the bar for publication.

**Award:**

No

---

### Decision · Program_Chairs · 2022-09-14

Reject